# Disease Resistance and Molecular Variations in Irradiation Induced Mutants of Two Pea Cultivars

**DOI:** 10.3390/ijms23158793

**Published:** 2022-08-08

**Authors:** Dong Deng, Suli Sun, Wenqi Wu, Chao Xiang, Canxing Duan, Dongmei Yu, Xuehong Wu, Zhendong Zhu

**Affiliations:** 1Institute of Crop Sciences, Chinese Academy of Agricultural Sciences, Beijing 100081, China; 2College of Plant Protection, China Agricultural University, Beijing 100193, China; 3Crop Research Institute, Sichuan Academy of Agricultural Sciences, Chengdu 610066, China

**Keywords:** *Pisum sativum*, mutant, disease resistance, genetic variation, *er1-2*

## Abstract

Induced mutation is useful for improving the disease resistance of various crops. Fusarium wilt and powdery mildew are two important diseases which severely influence pea production worldwide. In this study, we first evaluated Fusarium wilt and powdery mildew resistance of mutants derived from two elite vegetable pea cultivars, Shijiadacaiwan 1 (SJ1) and Chengwan 8 (CW8), respectively. Nine SJ1 and five CW8 M_3_ mutants showed resistant variations in Fusarium wilt, and the same five CW8 mutants in powdery mildew. These resistant variations were confirmed in M_4_ and M_5_ mutants as well. Then, we investigated the genetic variations and relationships of mutant lines using simple sequence repeat (SSR) markers. Among the nine effective SSR markers, the genetic diversity index and polymorphism information content (PIC) values were averaged at 0.55 and 0.46, which revealed considerable genetic variations in the mutants. The phylogenetic tree and population structure analyses divided the M_3_ mutants into two major groups at 0.62 genetic similarity (*K* = 2), which clearly separated the mutants of the two cultivars and indicated that a great genetic difference existed between the two mutant populations. Further, the two genetic groups were divided into five subgroups at 0.86 genetic similarity (*K* = 5) and each subgroup associated with resistant phenotypes of the mutants. Finally, the homologous *PsMLO1* cDNA of five CW8 mutants that gained resistance to powdery mildew was amplified and cloned. A 129 bp fragment deletion was found in the *PsMLO1* gene, which was in accord with *er1-2*. The findings provide important information on disease resistant and molecular variations of pea mutants, which is useful for pea production, new cultivar breeding, and the identification of resistance genes.

## 1. Introduction

Pea (*Pisum sativum* L.), including dry pea and green pea, is an important grain legume and vegetable for human food and animal feed in many countries [1,2,3]. The pea is regarded as an inexpensive and available source of proteins, minerals, vitamins, dietary fiber, and other nutrients, and can be used to formulate drugs to lower glycemic index and promote gastrointestinal digestion [3,4,5]. Up to now, the pea has been grown in more than 100 countries worldwide, with an annual cultivation area of more than 11 million hectares, ranking fourth in edible beans production after common bean, chickpea, and cowpea [6].

Biotic stresses have been the major yield-limiting factors of pea production worldwide. Among biotic stresses, Fusarium wilt, caused by *Fusarium oxysporum* f. sp. *pisi* (*Fop*) [7,8], and powdery mildew, induced by *Erysiphe pisi* D.C. [9,10], are the most important diseases in the pea, causing severe yield loss [7,9]. The deployment of resistant pea cultivars is considered to be the most effective and highly desirable method to manage these stubborn diseases, even though conventional disease control methods, such as rotation and the use of fungicides, may reduce the damage of these diseases [7,8,9,10]. The genetic mapping of resistance genes and the development of functional markers can facilitate and speed up the process of disease resistance breeding [11,12].

Generally, resistance sources that can be used for disease resistant breeding are mainly concentrated on a few backbone parents and their derived lines, which narrows the genetic basis of commercial cultivars [13,14]. It is easy for pathogens to overcome resistance in cultivars, if the cultivars with similar genetic background have been planted [15,16]. Thus, novel and diverse resistance germplasms used for breeding constantly need to be created and enriched through a variety of methods [17,18]. Among the breeding strategies, induced mutation plays an important role in the improvement, innovation, and utilization of germplasms due to its advantages of expanding the genetic spectrum and stable genetic progeny [18,19,20].

Induced mutation is an alternative for the improvement of desired traits in agricultural and horticultural crops. Pea is one of the first crops to be used for radiation mutagenesis research. In 1910, Schmidt gained larger plants following X-rays irradiation of soaked pea seeds [21]. Subsequently, the induced mutation of pea cultivars has been conducted in many pea-producing countries, the objectives of which were mainly focused on the agronomic traits of high yield, grain quality, and early maturity [22]. As a classic model plants for genetic research, the pea has also been used in mutation research for other agronomic traits, including compound inflorescence, compound leaves, zygomorphic flowers, and nodulation [23]. However, there have been only a few studies on mutagenesis of disease resistance in the pea so far. Gamma rays and ethyl methane sulfonate (EMS) were applied in a mutation induction of rust resistance for susceptible pea genotypes, and plants resistant to *U. pisi* were selected in the M_2_ and confirmed in the M_3_ [24]. Sharma et al. [8] screened out 25 pea mutants exhibiting complete or enhanced Fusarium wilt resistance using gamma rays or EMS mutagenesis. Pereira et al. [25] reported the first experimentally induced powdery mildew-resistant mutants of pea, which were obtained by the chemical mutagenesis of two susceptible commercial cultivars, Frilene and Solara, with the alkylating mutagenic compound ethylnitrosourea (ENU).

Induced mutations also play a key role in the genetic studies and identification of pea functional genes, such as genes regulating flowering [26], starch composition and use [27], symbiosis [28], and leaf development [29]. Powdery mildew resistance gene *er1* in the pea was first found in spontaneous mutation [30]; its recessive inheritance, durable, and broad-spectrum resistance characteristics are similar to those of barley powdery mildew resistance gene *MLO* [31]. Pavan et al. [32] developed a pea-induced mutant to identify the structure and function of pea powdery mildew resistance gene *er1* and revealed that *er1* resistance is associated with loss-of-function mutations at an *MLO* homologous locus *PsMLO1*. A novel allelic variation at the locus *PsMLO1* was also found in an ENU-induced powdery mildew resistant mutant pea line [33].

In China, previous studies of induced mutation in the pea were centered on the mutagenic effect of different mutagens, such as gamma rays, neutrons, non-vacuum pulsed electron beams, colchicine, dimethyl sulfoxide (DMSO), EMS, and sodium azido [34,35,36]. Generally, the emergence rate, chromosomal aberration rate, and yield of mutants were investigated to explore the suitable mutagenesis trigger and mutagenesis conditions [37]. Agronomic traits associated with yields, including plant height, leaf shape, branch number, and pod number [38] were evaluated in a few studies. Up until now, no reports were focused on disease resistance mutation of the pea in China.

The pea cultivars Shijiadacaiwan 1 (SJ1) and Chengwan 8 (CW8) are two elite vegetable pea cultivars bred by the Sichuan Academy of Agricultural Sciences which are resistant to Fusarium wilt [39], but susceptible to powdery mildew. The objective of this study was to identify phenotypic variations of resistance to Fusarium wilt and powdery mildew and genetic variations in mutants derived from SJ 1 and CW 8, select excellent resources, and identify a powdery mildew resistance gene for disease resistance breeding.

## 2. Results

### 2.1. Resistance Variation

#### 2.1.1. Fusarium Wilt

The SJ1, CW8, 45 M_3_ mutants (Y2-Y50) of SJ1, and 43 M_3_ mutants (Y152-Y198) of CW8 were evaluated for their resistance to the *Fop* isolate PF22b at 28 days post-inoculation (dpi) (Appendix A). A total of 15 mutants showed resistant variations (Table 1). All seedlings of nine SJ1 mutants: Y3, Y4, Y23, Y25, Y38, Y39, Y47, and Y48 (Figure 1A), and three CW8 mutants: Y160, Y192, and Y198 (Figure 1B), have become susceptible to Fusarium wilt, showing typical symptoms of yellowed leaves and dwarfed, withered, and dead plants; the disease index (DI) of these mutants was 100.00. The Y37 and Y186 were heterozygous for resistance to Fusarium wilt and produced two types of resistant and susceptible seedlings whose DI were between 35.00 and 75.00, showing that almost half of the seedlings were dead, while the others only showed slight or mild yellowing symptoms on the lower leaves. The SJ1, CW8, and remaining 72 mutants were symptomless or showed mild symptoms on the lower leaves, as indicated by a DI of less than 35.00, but more than 15.00.

The DI of 17 selected M_4_ lines derived from Y4, Y37, Y160, Y187, and Y192 to Fusarium wilt were consistent with that of their M_3_ mutants, respectively (Table 1). Fusarium wilt resistance or susceptibility of three M_4_ lines derived from Y186 were homozygous because their DIs were less than 35.00 or more than 75.00. Three M_4_ lines derived from Y37 were homozygous in resistance, with the DI reduced to 20.00, and five M_4_ lines acquired from Y198 had varying DI ranging from 20.00 to 76.00, with a mean DI of 44.00. All 40 selected M_5_ lines from M_4_ lines Y186-1, Y186-5, Y186-8, Y187-5, Y187-9, Y187-10, Y187-11, and Y187-12 maintained resistance to Fusarium wilt with DI of 20.00, while 5 M_5_ lines derived from M_4_ line Y186-2 were susceptible to Fusarium wilt, with DIs of more than 75.00 (Table 1).

#### 2.1.2. Powdery Mildew

Resistance of the 88 M_3_ mutants and 2 parents to *E. pisi* isolate EPYN was identified at the seedling stage (Appendix A). At 10 dpi, all seedlings of SJ1, CW8, 45 SJ1 mutants, and 38 CW8 mutants were heavily infected by *E. pisi* EPYN and covered by abundant mycelia and conidia, as indicated by IT 4. In contrast, all seedlings of Y192 and some seedlings of Y160, Y186, Y187, and Y198 were immune, and no symptoms were visible (Figure 1C, Table 1). Other seedlings of Y160 and Y198 showed little mycelia with IT 1, while those of Y186 and Y187 showed the same symptoms as their susceptible parents with IT 4 (Table 1).

Powdery mildew phenotypes of the selected M_4_ and M_5_ lines were also evaluated at the seedling stage (Table 1). Five M_4_ lines derived from Y4 and Y37 exhibited the same powdery mildew IT value (4) as their parents, while M_4_ lines Y160-1 and Y192-1 derived from Y160 and Y192, respectively, maintained the same immunity to powdery mildew as Y160 and Y192, with IT of 0. Similarly, five M_4_ lines obtained from Y198 were immune or resistant, since Y198 mixed with immune and resistant plants. Among nine M_4_ lines derived from heterozygous Y186 and Y187, seven from resistant plants of M_3_ parents were resistant lines, with IT 0 or IT1, and Y186-1 derived from a susceptible plant of Y186 was susceptible, with IT 4, while Y187-12 from a susceptible plant of Y187 was heterozygous, with ITs of 0 and 4. Moreover, similar resistant reactions were observed in M_5_ lines obtained from the selected M4 plants of Y186 and Y187 at the seedling stages (Table 1).

The Y160, Y186, Y187, Y192, Y198, and CW8 were inoculated with *E. pisi* EPYN at the adult plant stage under glasshouse conditions. Reactions to powdery mildew at adult plants of the selected M_4_ lines in the glasshouse were consistent with those at the seedling stage (Figure 1D, Table 2).

### 2.2. Molecular Variation Analysis of Mutants

Among the 11 simple sequence repeat (SSR) markers, the amplification products of nine SSR markers revealed polymorphisms between wild-type cultivars and mutants (Table 3). The nine markers produced 45 amplified fragments across SJ1, CW8, and their representative M_3_ mutants, with an average of 5 alleles, and the effective alleles varied from 1.54 (for 24407) to 3.56 (for EST709), with a mean value of 2.38. The values for major allele frequency ranged from 0.33 in primer EST709 to 0.79 in 24407, with an average of 0.54. The highest (0.72) and lowest (0.35) gene diversity indexes were obtained from the EST709 and EST921, respectively, and the average was 0.55. Likewise, 24407 (0.32) provided a smaller value than other markers in polymorphism information content (PIC), while the largest value of these indicators was provided by EST709 (0.67), with a mean value of 0.46.

A phylogenetic tree was constructed to elucidate the genetic relationships of mutants and parent cultivars (Figure 2). The genetic similarity between pea mutants and their relatives was presented in Figure 3. The two wild-type cultivars and their representative mutants were divided into two main groups at 0.62 genetic similarity, respectively. The SJ1 and its 10 mutants constituted Group I, whose similarity coefficient varied from 0.33 to 1.00, while 12 mutants with their wild-type CW8 were clustered in Group II, and their similarity coefficient was 0.33 to 1.00 as well. The smallest similarity coefficient among the members of Group I was 0.33, which was the largest similarity coefficient among the members of Group II. Group I could be further divided into three subgroups at 0.86 genetic similarity. Subgroup I-a only included the wild type SJ1; I-c contained two Fusarium wilt susceptible lines Y39 and Y47, and their similarity coefficient was 1.00; for I-b, three Fusarium wilt resistant lines: Y5, Y24, and Y37, and five Fusarium wilt susceptible lines: Y3, Y4, Y23, Y25, and Y46, were clustered in the independent clade, respectively, and the similarity coefficient varied between 0.89 and 1.00. The similarity coefficient between SJ1 and the mutants in subgroup I-b and I-c ranged from 0.44 to 0.67, and the similarity coefficient of subgroup I-b and I-c was 0.33 to 0.44. Group II was made up of two subclusters: one included seven Fusarium wilt resistant CW8 mutants: Y152, Y156, Y157, Y159, Y168, Y169, and Y175, and their wild type CW8, with the similarity coefficient of 1.00; the other contained five powdery mildew resistant CW8 mutants: Y160, Y186, Y187, Y192, and Y198, whose similarity coefficient varied from 0.67 to 1.00. However, the similarity coefficient range between Subgroup II-a and II-b was 0.33 to 0.56.

A population structure analysis was performed with a predefined number of groups (*K*) ranging from 1 to 10. The optimal *K* was determined using two methods, an ad-hoc statistic (*ΔK*), which was based on the rate of change in the log probability of the data between successive K-values, and the calculated likelihood value (ln*P*(*D*)), which was obtained from STRUCTURE software runs. The result of this calculation showed that when *K* = 2, Δ*K* reached its maximal value (Appendix A), which corresponded to a division of two genetic populations (Figure 2). Of the total 24 accessions, 11 were from Group I, and the other 13 were from Group II. The two genetic populations were consistent between the groups in phylogenetic analysis when the genetic similarity was 0.62. Meanwhile, structure simulation described that the average of ln*P*(*D*) against *K* = 5 was addressed to be the best *K* (Appendix A), which indicated that 5 subgroups could include all of the 24 accessions with the highest probability (Figure 2). Each accession was assigned to these subgroups according to its genotype. The estimated population structure of *K* = 5 suggested that genotypes with partial membership exhibited distinctive identities, which fitted with the phylogenetic analysis of subgroups when the genetic similarity was 0.86.

A dendrogram constructed using the unweighted pair group method with arithmetic mean (UPGMA), based genetic analysis divided CW8 and its M_3_ and M_4_ mutants into three major clusters with the 0.87 similarity coefficients (Figure 4). The genetic analysis results of M_3_ mutants were similar to those in Figure 2. Y186 M_3_ and M_4_ mutants comprised Cluster B, and Y186 and Y198 formed a subcluster with Y186-1, Y186-5, and Y186-12. Equally, Y160, Y187, and Y192 made up Cluster C with Y187 M_4_ mutants. Five Y187 M_4_ mutants, Y187-2, Y187-3, Y187-6, Y187-7, and Y187-9, assembled in the same clade with Y187; and two, Y187-8, Y187-10 assembled with Y160 and Y192.

### 2.3. PsMLO1 Sequence Analysis

Powdery mildew phenotypes in the M_4_ and M_5_ lines derived from Y186 and Y87 revealed that powdery mildew resistance in some lines should be genetically recessive, because resistant lines were produced from both resistant and susceptible plants. Since the powdery mildew susceptible gene *PsMLO*1 in the pea can be naturally and artificially mutated into a recessive disease resistance gene *er1* [9,10,33], we analyzed *PsMLO*1 cDNA sequences of five powdery mildew resistant mutants and their parent CW8. Based on ten clones, the *PsMLO*1 cDNA sequences of CW8, the susceptible wild-type cultivar, were identical to those of *PsMLO1* from the pea cultivar Sprinter (NCBI accession number: FJ463618). In contrast, the *PsMLO1* cDNA of the five resistance M_3_ mutants, Y160, Y186, Y187, Y192, and Y198, were different from CW8 and Sprinter. Sequence alignment analysis indicated that the five mutants *PsMLO1* cDNA had a 129-bp deletion between positions 1171 and 1299, which were exons 13 and 14 of the *PsMLO1* gene (Figure 5). The fragment deletion mutation has been reported in some pea accessions, which was recognized as *er1-2* [10,33].

## 3. Discussion

The induced mutation is a significant approach to crop germplasm innovation and breeding, as well as the main technical mean of gene discovery and target identification in crop plants [40,41]. Common mutation breeding strategies include gamma rays and X-rays using radiation [8,21,36]; EMS, ENU, DMSO, and colchicine using chemicals [25,33,35]; Fok1 and CRISPR/Cas9 using enzymes [42]; and microgravity and cosmic radiation using space-breeding [43,44]. To date, more than 3200 mutagenized plant cultivars have been released, most of which were direct mutants of crop plants [40].

The pea diseases, especially Fusarium wilt and powdery mildew, are major factors limiting yield in production [8,25]. Thus, we screened M_3_ mutants derived from two elite vegetable pea cultivars, SJ1 and CW8, with resistance to Fusarium wilt and powdery mildew as the main traits in this study. Despite being resistant to Fusarium wilt in SJ1 and CW8, nine SJ1 and five CW8 M_3_ mutants converted to disease susceptible or heterozygous lines. Both parents were susceptible to powdery mildew, and five mutants of CW8 which were resistant or heterozygous to this disease were screened, but no resistant or heterozygous mutants of SJ1 were detected. Sharma et al. [8] obtained some pea mutants with resistance or improved resistance to Fusarium wilt by the radiation mutagenesis of two susceptible genotype cultivars. The results of Sharma et al. [8] and our research proved that radiation mutagenesis could effectively induce the resistance variation of pea Fusarium wilt and powdery mildew, which is an effective means to create pea disease-resistant germplasms and identify resistant genes. Among the five powdery mildew resistant M_3_ lines screened in CW8, the Y160, Y192, and Y198 lost Fusarium wilt resistance, but some plants of Y186 and Y187 retained this trait. Pure Fusarium wilt and powdery mildew resistant lines, such as Y186-5, Y186-8, Y187-9, Y187-10, Y187-11, Y187-12-6, and Y187-12-7, were obtained through further selection in the M_4_ and M_5_ lines, which would be of great value to pea production and breeding. Unfortunately, mutant lines resistant to powdery mildew could not be screened from SJ1 mutants. The explanations might be that the number of M_3_ mutants was small, or the radiosensitivity of this cultivar was weaker than that of CW8. The difference in radiosensitivity, influenced by genetics, existed in various cultivars of the same crop [45,46].

Genetic variation levels of mutants can be evaluated by various molecular markers [47,48,49]. Compared with other genetic markers, SSR markers have significant potential in distinguishing pea genotypes [13,50,51]. The PIC is a principal pointer to discriminate the polymorphism percentage of one marker at a precise locus; and the PIC values in SSR markers are positively correlated with percent polymorphism [52,53]. In addition, SSR markers with PIC values greater than 0.50 are considered as to be effective in discriminating the polymorphism rate, PIC values between 0.25 to 0.50 are considered to be intermediate, and PIC values less than 0.25 are considered to be low [54]. In this study, nine effective SSR markers amplified one to four alleles in all genotypes, and the mean PIC value was 0.46, which indicated all tested mutants have exhibited considerable levels of genetic diversity. These findings supported mutagenesis as an effective means to enrich the genetic diversity of crops, especially in a largely self-pollinating pea [13,55,56].

The UPGMA clustering analysis, in agreement with population structure, delineated all 24 accessions in two groups at 0.62 genetic similarity (*K* = 2, Figure 2). The greatest genetic variation was displayed between different pea mutants in two groups. Thus, the mutants, generated from a cultivar, were obviously grouped together in the phylogenetic tree. Five subgroups were further divided when the *K* = 5, with 0.86 genetic similarity (Figure 2). Interestingly, these subgroups were associated with disease-resistance. Subgroup I-a and I-b in group I were resistant or heterozygous to Fusarium wilt, while I-c and I-b were susceptible. Similar analyses were performed on Group II, where Subgroup II-a displayed resistance to Fusarium wilt, but susceptibility to powdery mildew, whereas II-b showed inverse results, with susceptibility to Fusarium wilt, but resistance or heterozygosity to powdery mildew. Associations of phenotype variations with molecular markers in crop mutants have been reported. Theerawitaya et al. [57] used AFLP to investigate genetic variations associated with salt tolerance in mutants of KDML105 rice, and the mutants were clustered into 3 different groups containing different salt tolerance characteristics. Ramchander et al. [49] detected marker trait associations in gamma irradiated mutants of rice and found an SSR marker strongly associated with the traits of plant height, panicle length, and the number of grains per panicle. However, the association of molecular markers with phenotypes of mutants requires further validation by linkage analysis and genetic mapping.

Among the three identified pea powdery mildew resistance loci, *er1*, *er2,* and *Er3*, the *er1* gene was widely used because of its recessive inheritance, broad spectrum, and persistence [9,58,59]. The resistance gene *er1* was conferred by loss of function mutations in susceptible gene *PsMLO1* [9]. To date, 12 *er1* genes have been identified so far, including two artificial chemical mutations (*er 1-5* and *er 1-10*) and ten natural mutations (the rest of the *er1* genes) [32,33]. Among these *er1* genes, *er1-1* (*er1mut1*), *er1-5*, *er1-6*, and *er1-10* were single base mutations; *er1-3*, *er1-4*, and *er1-9* had single base deletions; *er1-12* showed single base insertion; *er1-7* and *er1-8* held fragment deletions; and *er1-2* exhibited large transposon insertion or a deletion of unknown size in the 13–14th exons [10,12,58,59]. Pereira et al. [25] treated powdery mildew susceptible cultivar Solara with ENU to induce the 680th base of *PsMLO1* gene cDNA from C to G, which was the same as the natural mutation *er1-1* [32,33,58]. Chemical mutagenesis mainly caused base substitution by DNA base alkylation, while gamma irradiation produced reactive oxygen species (ROS), which cause base substitutions, insertions, deletions, inversions and translocations [60,61]. In this study, we analyzed the cDNA sequences of the homologous *PsMLO1* gene of five CW8 mutants resistant to powdery mildew and revealed that these mutants had a 129-bp deletion between positions 1171 and 1299 in the *PsMLO1* gene. This result was consistent with the *er1-2* mutation in the powdery mildew resistant cultivars, Stratagem, Xucai 1, Yunwan 21, and Yunwan 23 [10,58]. The pea genome contains a large proportion of transposable elements (TEs), and the high abundance of LTR/Gypsy Ogre TEs likely influenced the pea genome’s rapid evolution [62]. The activity of transposable elements could be activated by a variety of stresses, such as gamma irradiation [63,64,65]. Hung et al. [66] applied a transposon-based marker system to reveal abundant dynamic and active mobility levels of transposons in gamma-ray irradiated soybean mutant lines. Sen et al. [67] detected retrotransposon insertional variations in drought-tolerant wheat mutants obtained by gamma ray irradiation and found high polymorphisms of retrotransposons microsatellite amplified polymorphism (REMAP) markers and inter-retrotransposon amplified polymorphism (IRAP) markers. The *b* gene of the pea encodes a defective flavonoid 3′,5′-hydroxylase, and confers pink flower color [68]. Moreau et al. [68] found allelic pink-flowered pea *b* mutant lines generated by fast neutron bombardment that carried a variety of lesions in the gene, including complete gene deletions, and suggested the action of a nearby transposon activated in the FN mutagenesis may be prone to deletion.

The induced mutation is an important means to create new germplasm and broaden the genetic basis of commercial crop cultivars [18,40]. In this study, the resistance of M_3_-M_5_ mutants derived from two pea cultivars, SJ1 and CW8, to Fusarium wilt and powdery mildew were evaluated, and genetic variations in the mutants were analyzed by using SSR markers. In addition, the *PsMLO1* allele of powdery mildew resistant mutants was identified as well. These results indicated that induced mutation has led to significant disease resistance and genetic variation in pea SJ1 and CW8 mutants. The resistant mutants can be used for pea production, the development of new resistant pea cultivars, and the identification of the resistance genes.

## 4. Materials and Methods

### 4.1. Pea Mutants and Pathogen Isolates

The pea wild type cultivars Shijiadacaiwan 1 (SJ1), Chengwan 8 (CW8), and some of their M_3_ mutants (Y2-Y50 from SJ1, Y152-Y198 from CW8) were provided by the Crop Research Institute, Sichuan Academy of Agricultural Sciences (Appendix A). The mutants were induced by ^60^Co γ-ray at the doses of 170 Gy (1 Gy/min) [36]. The *F*. *oxysporum* f. sp. *pisi* race 5 isolate PF22b, collected in Sichuan Province, China [39], and the *E. pisi* isolate EPYN from Yunnan Province, China [10], were used for resistance evaluation. The isolate PF22b was stored at −80 °C, and EPYN was maintained through continuous re-inoculation of seedlings of the pea cultivar Longwan 1.

### 4.2. Resistance Evaluation

#### 4.2.1. Fusarium Wilt

Eighteen seeds of each mutant and wild type were planted in three duplicate paper cups (600 mL) filled with fresh vermiculite, and the planted cups were placed in the greenhouse at 18–22 °C for two weeks. The inoculum of Fusarium wilt isolate PF22b was prepared by placing several mycelial plugs (3–4 mm in diameter) into 100 mL pea soup (peas 40 g, distilled water 1 L, boiled for 60 min, filtered, sterilized at 121 °C for 30 min), which was incubated for 2 days in an incubation shaker (27 °C, 100 rpm) [39,69]. After filtering through four layers of gauze, the conidial suspension was adjusted to a final concentration of 1.0 × 10^7^ spores/mL to inoculate the plants. The 14-day-old seedlings were uprooted, and the roots were washed thoroughly under running tap water. The bottom 1/3 sections of the plant roots were removed, and the trimmed seedlings were dipped in the spore suspensions for 3 min and then transplanted into a new cup [39,70]. Inoculated seedlings were cultured in a greenhouse at 27 ± 2 °C with natural light. The percentage of leaves showing symptoms for each individual plant (PSL) was used to estimate the symptom rate at the leaf level with a 0–5 scale [39]: 0, PSL = 0; 1, 0 < PSL ≤ 25%; 2, 25% < PSL ≤ 50%; 3, 50% < PSL ≤ 75%; 4, 75% < PSL < 100%; 5, PSL = 100%. These data were used to calculate the disease index (DI) for each cultivar by using the formula: DI = [∑ (n × s)/(N × 5)] ×100, where n = the number of plants at that grade, s = the scale of the disease severity, and N = the total number of plants tested. The evaluating resistance to *Fop* race 5 was defined according Deng et al. [39], with slight modifications: highly resistant (HR), 0 ≤ DI ≤ 15; resistant (R), 15 < DI ≤ 35; moderate (M), 35 < DI < 70; susceptible (S), 70 ≤ DI < 90; highly susceptible (HS), 90 ≤ DI ≤ 100. For those mutants identified as highly resistant or resistant to *Fop* race 5, repeated identification was performed.

#### 4.2.2. Powdery Mildew

The growth of plants was evaluated in the same way as the resistance for Fusarium wilt. The powdery mildew inoculation was performed by shaking off fresh conidia from heavily infected Longwan 1 plants onto the tested seedlings [10]. Inoculated seedlings were cultured in a glasshouse at 20 ± 2 °C. The disease severity was determined on 0–4 infection types (IT) at 10 days post-inoculation (dpi), according to the methods of Sun et al. [10]: IT 0, seedlings with no visible symptoms; IT 1, seedlings with necrotic reaction and no or little mycelial development; IT 2, seedlings with necrotic reaction and moderate mycelial development; IT 3, seedlings with moderate mycelial development and little sporulation; IT 4, seedlings with abundant mycelial development and profuse sporulation. Plants with a score of 0 were considered *E. pisi*-immune, while those with scores of 1, 2, 3, and 4 were considered as *E. pisi*-resistant and *E. pisi*-susceptible, respectively. Those mutants with variations in resistance to *E. pisi* were retested.

### 4.3. Molecular Variation Analysis of Mutants

The genomic DNA of SJ1, CW8, their representative M_3_ mutants (Y3, Y4, Y5, Y23, Y24, Y25, Y37, Y39, Y46, and Y47 derived from SJ1; Y152, Y156, Y157, Y159, Y160, Y168, Y169, Y175, Y186, Y187, Y192, and Y198 derived from CW8), and the Y186 and Y187 M_4_ lines were extracted from young leaves using the DNAsecure Plant Kit (Tiangen Biotech, Beijing, China), according to the manufacturer’s instruction. The DNA solution was diluted and stored at −20 °C until use. Eleven SSR markers (Appendix A) were used to evaluate genetic variations among SJ1, CW8, and their mutants [71,72]. Polymerase chain reactions (PCR) were carried out using a Gene Amp 9700 thermocycler (Applied Biosystems, Foster City, CA, USA) in 10 μL reaction mixtures containing: 10 ng of DNA, 0.4 μL of each primer, 5 μL of 2 × *Taq* PCR Master Mix (Tiangen Biotech, Beijing, China), and 3.2 μL ddH_2_O. The PCR program was as follows: 94 °C for 5 min; then 35 cycles of denaturation at 94 °C for 30 s, annealing at 51–63 °C for 30 s, and extension at 72 °C for 1 min; and a final extension at 72 °C for 10 min. The PCR products were separated on a 6% polyacrylamide gel with 1 × TBE and stained with silver nitrate solution and formaldehyde.

The number and frequency of alleles were counted based on the SSR electrophoresis pattern at each locus for each sample, according to the difference in band size of the microsatellite amplification products of SJ1, CW8, and the different mutants. The number of genotypes, major allele frequency of each SSR marker, and genetic similarity of each line were computed using the PopGene 32 software (University of Alberta, Edmonton, AB, Canada) [73], and the results of genetic similarity were transformed as a heatmap in Rstudio 4.2.1 (RStudio, Inc., Boston, MA, USA). The PowerMarker 3.25 software (North Carolina State University, Raleigh, NC, USA) was used for calculating the gene diversity index and the polymorphism information content [74]. Phylogenetic trees were constructed using the UPGMA algorithm in the NTSYSpc 2.1 software (Exeter Software, Setauket, NY, USA) [75]. To analyze population structure, a Bayesian population analysis was performed in STRUCTURE 2.3.4 (The Pritchard Lab, Stanford University, Stanford, CA, USA) [76,77], with the parameter of “admixture model” with a burn-in period of 10,000 followed by 10,000 iterations. Five independent runs were calculated at each *K* level with a range from 1 to 10. The likelihood value of data (ln*P*(*D*)) [76] and calculation of Delta K (Δ*K*) [78] were used to evaluate the optimum value for *K*.

### 4.4. RNA Extraction and PsMLO1 Sequence Analysis

The extraction of total RNA and the synthesis of cDNA from wild-type CW8 and its powdery mildew resistant M_3_ mutants were performed using the RNAprep Pure Plant kit (Tiangen Biotech, Beijing, China) and BioRT Two Step RT-PCR kit (Hangzhou Bioer Technology, Hangzhou, China), respectively, according to the manufacturer’s instruction.

To identify the resistance alleles at the *er1* loci, the full-length cDNAs of the *PsMLO1* homologs were amplified using the *PsMLO1*-specific primers (PsMLO1F: 5′-AAAATGGCTGAAGAGGGAGTT-3′; PsMLO1R: 5′-TCCACAAATCAAGCTGCTACC-3′) [10,32]. Except for the use of 5 times the reaction mixtures and 58 °C annealing temperature, the PCR reaction and amplification were the same as for the genetic analysis. The PCR products were purified with TIANquick Midi Purification Kit (Tiangen Biotech, Beijing, China). The cloning of the PCR products was completed using the pEasy-T5 vector (TransGen Biotech, Beijing, China), and the sequencing reactions of 10 clones per mutant and wild-type parent were performed by the Sangon Biotech Co., Ltd. (Shanghai, China). The resulting sequences were aligned and analyzed with pea cultivar Sprinter (NCBI accession number: FJ463618) using DNAMAN 6.0 (Lynnon Biosoft, PQ, Canada).

## 5. Conclusions

In this study, we first evaluated the resistance of the SJ1 and CW8 mutants to Fusarium wilt and powdery mildew, and found the nine SJ1 (screened for Fusarium wilt) and five CW8 M_3_ mutants (screened for Fusarium wilt and powdery mildew) with resistant variations. Then, SSR markers were used to detect genetic diversity in the mutants, and significant molecular variations associated with resistance to Fusarium wilt and powdery mildew were revealed in the mutants. Finally, we discovered that the mutants acquired resistance to powdery mildew due to a 129 bp fragment deletion of the *PsMLO1* gene, which was confirmed as *er1-2*. This study provided important information on disease resistant and molecular variations of pea mutants, which will be useful for pea production, new cultivar breeding, and the identification of resistance genes.

## Figures and Tables

**Figure 1 ijms-23-08793-f001:**
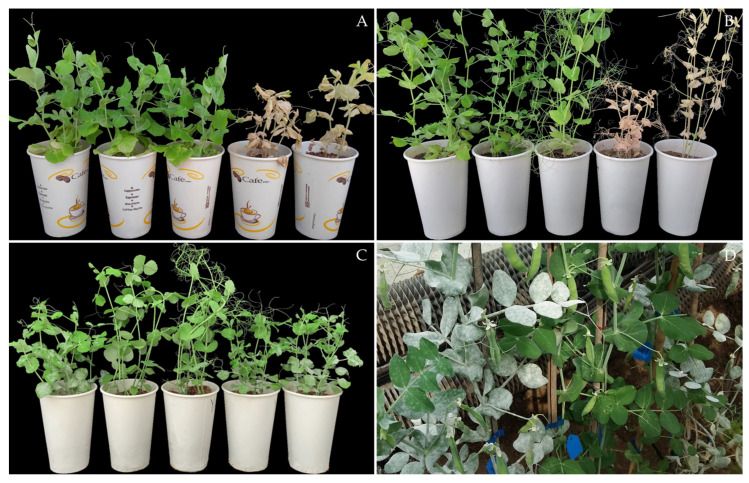
Disease resistant variations in M_3_ mutants: (**A**) reactions of Shijiadacaiwan 1 (on the left) and partial mutants to Fusarium wilt at seedling stage; (**B**,**C**) reactions of Chengwan 8 (on the left) and partial mutants to Fusarium wilt (**B**) and powdery mildew (**C**) at seedling stage; (**D**) reactions of Y187 partial mutant plants to powdery mildew at adult stage.

**Figure 2 ijms-23-08793-f002:**
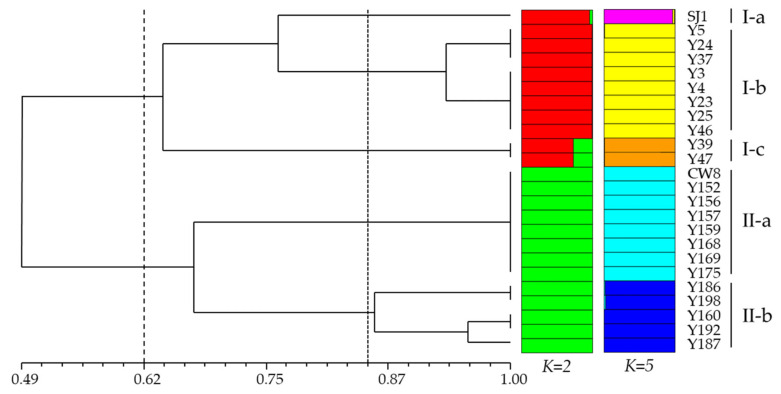
Phylogenetic tree revealed by the unweighted pair group method with arithmetic mean (UPGMA) molecular variation analysis and the population structure of pea M_3_ mutants and their origins based on SSR markers. SJ1, Shijiadacaiwan 1; CW8, Chengwan 8.

**Figure 3 ijms-23-08793-f003:**
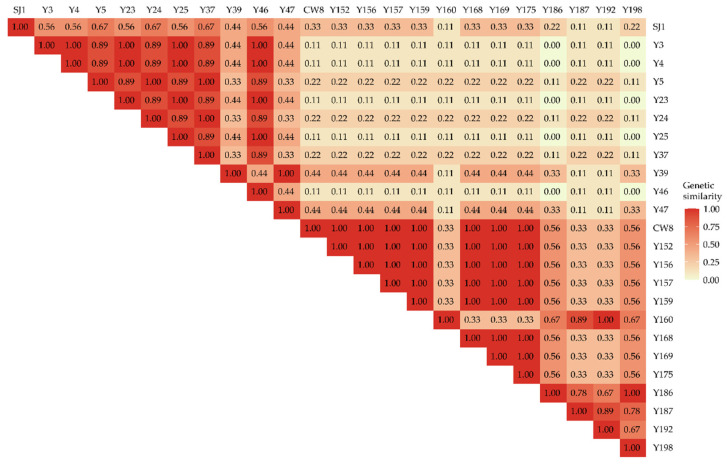
Genetic similarity of pea M_3_ mutants and their origins. SJ1, Shijiadacaiwan 1; CW8, Chengwan 8.

**Figure 4 ijms-23-08793-f004:**
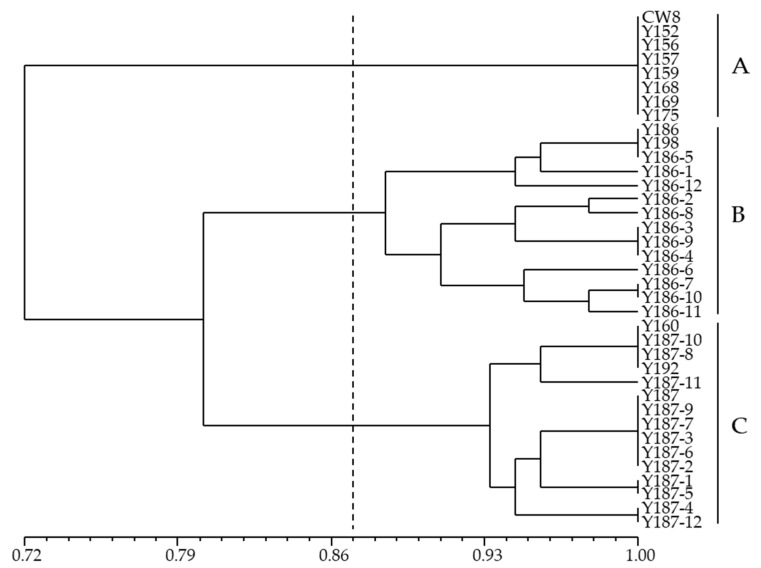
Dendrogram revealed using the unweighted pair group method with arithmetic mean (UPGMA) genetic analysis of CW8 and its mutants, based on SSR markers. CW8, Chengwan 8.

**Figure 5 ijms-23-08793-f005:**
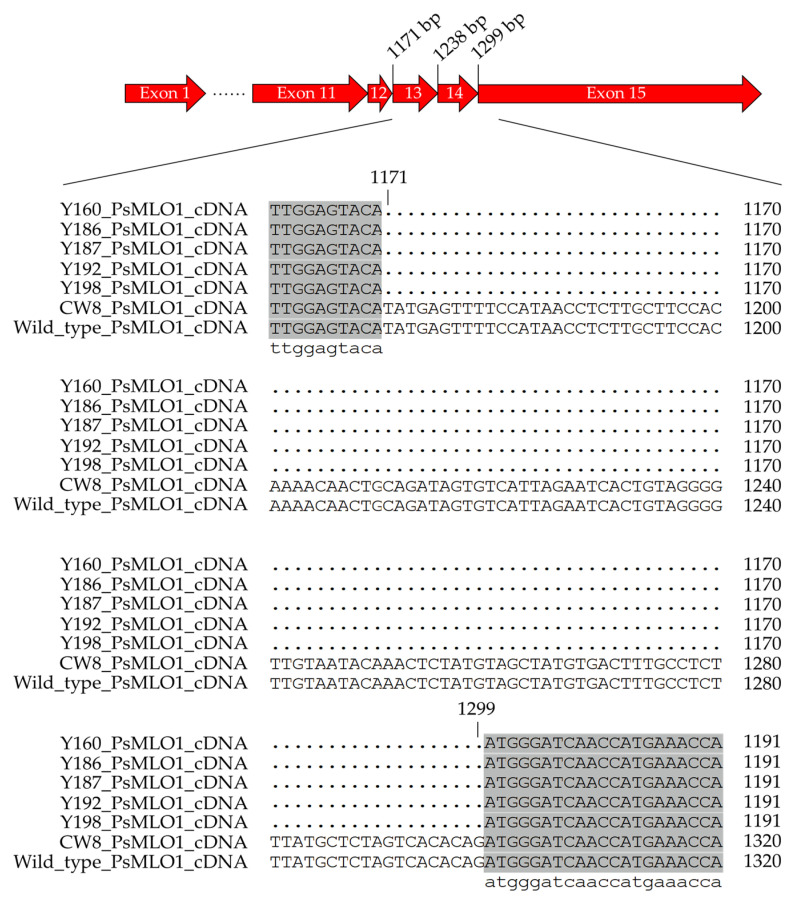
Comparison of *PsMLO1* cDNA sequence from powdery mildew resistant mutants, susceptible Chengwan 8, and wild-type Sprinter.

**Table 1 ijms-23-08793-t001:** Resistant reactions to Fusarium wilt and powdery mildew of mutants derived from pea cultivars Shijiadacaiwan 1 and Chengwan 8 at seedling stage.

Line ^1^	Parent	Generation	Fusarium Wilt		Powdery Mildew	
			Disease Index	Reaction ^2^	Infection Type	Reaction ^3^
SJ1	-	-	20.00	R	4	S
CW8	-	-	20.00	R	4	S
Y3	SJ1	M_3_	100.00	HS	4	S
Y4	SJ1	M_3_	100.00	HS	4	S
Y23	SJ1	M_3_	100.00	HS	4	S
Y25	SJ1	M_3_	100.00	HS	4	S
Y37	SJ1	M_3_	40.00	M	4	S
Y38	SJ1	M_3_	100.00	HS	4	S
Y39	SJ1	M_3_	100.00	HS	4	S
Y46	SJ1	M_3_	100.00	HS	4	S
Y47	SJ1	M_3_	100.00	HS	4	S
Y160	CW8	M_3_	100.00	HS	0, 1	I, R
Y186	CW8	M_3_	66.67	M	0, 4	I, S
Y187	CW8	M_3_	20.00	R	0, 4	I, S
Y192	CW8	M_3_	100.00	HS	0	I
Y198	CW8	M_3_	86.67	S	0, 1	I, R
Y4-1	Y4	M_4_	100.00	HS	4	S
Y4-2	Y4	M_4_	100.00	HS	4	S
Y37-1	Y37	M_4_	20.00	R	4	S
Y37-2	Y37	M_4_	20.00	R	4	S
Y37-5	Y37	M_4_	20.00	R	4	S
Y160-1	Y160	M_4_	100.00	HS	0	I
Y186-1	Y186	M_4_	20.00	R	4	S
Y186-2	Y186	M_4_	96.67	HS	0, 1	I, R
Y186-5	Y186	M_4_	20.00	R	0	I
Y186-8	Y186	M_4_	20.00	R	0	I
Y187-5	Y187	M_4_	20.00	R	0	I
Y187-9	Y187	M_4_	20.00	R	0	I
Y187-10	Y187	M_4_	20.00	R	0	I
Y187-11	Y187	M_4_	20.00	R	0	I
Y187-12	Y187	M_4_	20.00	R	0, 4	I, S
Y192-1	Y192	M_4_	100.00	HS	0	I
Y198-1	Y198	M_4_	36.00	M	0	I
Y198-2	Y198	M_4_	36.00	M	0	I
Y198-3	Y198	M_4_	52.00	M	0	I
Y198-4	Y198	M_4_	20.00	R	1	R
Y198-5	Y198	M_4_	76.00	S	0	I
Y186-1-1	Y186-1	M_5_	20.00	R	4	S
Y186-1-2	Y186-1	M_5_	20.00	R	4	S
Y186-1-4	Y186-1	M_5_	20.00	R	4	S
Y186-1-6	Y186-1	M_5_	20.00	R	4	S
Y186-1-8	Y186-1	M_5_	20.00	R	4	S
Y186-2-1	Y186-2	M_5_	100.00	HS	1	R
Y186-2-2	Y186-2	M_5_	88.00	S	1	R
Y186-2-3	Y186-2	M_5_	100.00	HS	0	I
Y186-2-4	Y186-2	M_5_	100.00	HS	0	I
Y186-2-6	Y186-2	M_5_	100.00	HS	1	R
Y186-5-2	Y186-5	M_5_	20.00	R	0	I
Y186-5-3	Y186-5	M_5_	20.00	R	0	I
Y186-5-4	Y186-5	M_5_	20.00	R	0	I
Y186-5-5	Y186-5	M_5_	20.00	R	0	I
Y186-5-6	Y186-5	M_5_	20.00	R	0	I
Y186-8-2	Y186-8	M_5_	20.00	R	0	I
Y186-8-4	Y186-8	M_5_	20.00	R	0	I
Y186-8-5	Y186-8	M_5_	20.00	R	0	I
Y186-8-6	Y186-8	M_5_	20.00	R	0	I
Y186-8-7	Y186-8	M_5_	20.00	R	0	I
Y187-5-1	Y187-5	M_5_	20.00	R	4	S
Y187-5-4	Y187-5	M_5_	20.00	R	4	S
Y187-5-5	Y187-5	M_5_	20.00	R	4	S
Y187-5-6	Y187-5	M_5_	20.00	R	4	S
Y187-5-8	Y187-5	M_5_	20.00	R	4	S
Y187-9-1	Y187-9	M_5_	20.00	R	0	I
Y187-9-2	Y187-9	M_5_	20.00	R	0	I
Y187-9-3	Y187-9	M_5_	20.00	R	0	I
Y187-9-4	Y187-9	M_5_	20.00	R	0	I
Y187-9-5	Y187-9	M_5_	20.00	R	0	I
Y187-10-2	Y187-10	M_5_	20.00	R	0	I
Y187-10-3	Y187-10	M_5_	20.00	R	0	I
Y187-10-4	Y187-10	M_5_	20.00	R	0	I
Y187-10-5	Y187-10	M_5_	20.00	R	0	I
Y187-10-6	Y187-10	M_5_	20.00	R	0	I
Y187-11-2	Y187-11	M_5_	20.00	R	0	I
Y187-11-3	Y187-11	M_5_	20.00	R	0	I
Y187-11-4	Y187-11	M_5_	20.00	R	0	I
Y187-11-7	Y187-11	M_5_	20.00	R	0	I
Y187-11-1	Y187-11	M_5_	20.00	R	0	I
Y187-12-1	Y187-12	M_5_	20.00	R	0, 4	I, S
Y187-12-2	Y187-12	M_5_	20.00	R	0, 4	I, S
Y187-12-3	Y187-12	M_5_	20.00	R	0, 4	I, S
Y187-12-6	Y187-12	M_5_	20.00	R	0	I
Y187-12-7	Y187-12	M_5_	20.00	R	0	I

^1^ SJ1, Shijiadacaiwan 1; CW8, Chengwan 8; ^2^ HR, highly resistant; R, resistant; M, moderate; S, susceptible; HS, highly susceptible. ^3^ I, immune; R, resistant; S, susceptible.

**Table 2 ijms-23-08793-t002:** Resistance reactions of mutants derived from pea cultivars Shijiadacaiwan 1 and Chengwan 8 to powdery mildew at the adult plant stage.

Lines ^1^	Parent	Generation	Powdery Mildew	
			Infection Type	Reaction ^2^
SJ1	-	-	4	S
CW8	-	-	4	S
Y4	SJ1	M_3_	4	S
Y37	SJ1	M_3_	4	S
Y160	CW8	M_3_	0	I
Y186	CW8	M_3_	0, 4	I, S
Y187	CW8	M_3_	0, 4	I, S
Y192	CW8	M_3_	0	I
Y198	CW8	M_3_	0	I
Y160-1	Y160	M_4_	0	I
Y186-1	Y186	M_4_	4	S
Y186-2	Y186	M_4_	0	I
Y186-5	Y186	M_4_	0	I
Y186-8	Y186	M_4_	0	I
Y187-9	Y187	M_4_	0	I
Y187-10	Y187	M_4_	0	I
Y187-11	Y187	M_4_	0	I
Y187-12	Y187	M_4_	0, 4	I, S
Y192-1	Y192	M_4_	0	I
Y198-1	Y198	M_4_	0	I
Y198-2	Y198	M_4_	0	I

^1^ SJ1, Shijiadacaiwan 1; CW8, Chengwan 8; ^2^ I, immune; R, resistant; S, susceptible.

**Table 3 ijms-23-08793-t003:** Genetic information of nine SSR markers on wild-type cultivars and mutants.

Marker	Na ^1^	Ne	MAF	H	PIC
25986	6	3.03	0.42	0.67	0.61
26117	3	1.99	0.54	0.50	0.37
25433	5	1.88	0.63	0.47	0.36
EST921	3	1.70	0.71	0.41	0.33
EST709	7	3.56	0.33	0.72	0.67
PSGAPA1	6	2.46	0.50	0.59	0.51
24407	4	1.54	0.79	0.35	0.32
24575	7	3.31	0.38	0.70	0.64
AD147	4	1.99	0.54	0.50	0.37
Mean	5	2.38	0.54	0.55	0.46

^1^ Na, Observed number of alleles; Ne, Effective number of alleles; MAF, Major allele frequency; H, gene diversity index; PIC, polymorphism information content.

## Data Availability

The data presented in this study are available on request from the corresponding author.

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
