# Peer review of "Disease Resistance and Molecular Variations in Irradiation Induced Mutants of Two Pea Cultivars"

_ijms, 2022, doi:10.3390/ijms23158793_

Round 1

Reviewer 1 Report

In this manuscript, Deng et al. present the resistance variations of two pea cultivars and their mutant derivatives over two fungal plant diseases. They also associate these phenotypes with the genetic affiliations of the derived mutants by using SSR typing and finally, they identify a deletion at the PsMLO1 gene that grants resistance to one of the diseases.

The paper is well written with a detailed introduction and a meticulous analysis. The discussion is thorough and transparent and the methods are appropriate.

In the results, Table 3 would be easier to read if it was given as a heatmap.

Please find some specific comments below:

L. 26: Please replace comma (",") with "that" or equivalent.

L. 27: Pls, replace "were" with "was".

L. 28: Pls, replace "provided" with "provide".

L. 29: Pls, replace "was" with "is".

L. 35: Pls, replace "vegetables" with "vegetable".

Author Response

Response to Reviewer 1 Comments

Thank you very much for providing much valuable and constructive comments and suggestions. We felt all of the comments provided were highly helpful for revising and improving this manuscript. We have studied the comments and suggestions carefully and revised our manuscript in detail according to the comments and suggestions that you have provided. The revisions were marked in red color for Reviewer 1 in this revised manuscript.

Major comments:

Point 1: In the results, Table 3 would be easier to read if it was given as a heatmap.

Response 1: Thanks for your good comment. We modified that table into a heatmap (Figure 3).

Minor comments.

Point 2: Please find some specific comments below:

L. 26: Please replace comma (",") with "that" or equivalent.

L. 27: Pls, replace "were" with "was".

L. 28: Pls, replace "provided" with "provide".

L. 29: Pls, replace "was" with "is".

L. 35: Pls, replace "vegetables" with "vegetable".

Response 2: Thanks very much for your careful and professional suggestions. We are sorry for our carelessness. We changed these improper words and marked in red.

Reviewer 2 Report

The paper provides good Results about molecular genetics of mutagenic products, the analysis of plant resistance is consistent.

γ techniques used to produce the plant material, even if used again in recent years, appears anachronic.

Conclusions can be improved.

Minor language issues are present.

Author Response

Response to Reviewer 2 Comments

Thank you very much for providing much valuable and constructive comments and suggestions. We felt all of the comments provided were highly helpful for revising and improving this manuscript. We have studied the comments and suggestions carefully and revised our manuscript in detail according to the comments and suggestions that you have provided. The revisions were marked in bule color for Reviewer 2 in this revised manuscript.

Major comments:

Point 1: γ techniques used to produce the plant material, even if used again in recent years, appears anachronic.

Response 1: Thanks for your good comment. We are considering other methods of mutagenesis breeding, such as EMS and CRISPR/Cas9.

Point 2: Conclusions can be improved.

Point 3: Minor language issues are present.

Response 2: Thanks very much for your professional suggestions. We improved our conclusions and made some modifications to minor language issues by blue color in this revised manuscript.